# Changes in Texture Characteristics and Special Requirements of Sichuan-Style Braised Beef for Industrial Production: Based on the Changes in Protein and Lipid of Beef

**DOI:** 10.3390/foods12071386

**Published:** 2023-03-24

**Authors:** Xiaoli Pu, Jinggang Ruan, Zhicheng Wu, Yong Tang, Ping Liu, Dong Zhang, Hongjun Li

**Affiliations:** 1College of Food and Bioengineering, Xihua University, Chengdu 610039, China; 2Chongqing Key Laboratory of Speciality Food Co-Built by Sichuan and Chongqing, Chengdu 610039, China; 3Food Industry Collaborative Innovation Center, Xihua University, Chengdu 610039, China; 4College of Food Science, Southwest University, Chongqing 400715, China

**Keywords:** stewing time, Sichuan-style braised beef, texture properties, oxidation, muscle fibers

## Abstract

This study aimed to investigate the optimal stewing time (0, 30, 60, 90, 120, and 150 min) for industrialized preparation of Sichuan-style braised beef with different demands. With prolonged stewing time, the hardness and chewiness of the braised beef initially increased and then decreased (*p* < 0.05), whereas springiness and cohesiveness gradually decreased. The moisture content of braised beef and the endogenous fluorescence intensity of braised beef protein significantly decreased (*p* < 0.05). However, the thiobarbituric acid reaction substances (TBARS) value and protein carbonyl content of braised beef greatly increased (*p* < 0.05). During the stewing process, the texture properties of Sichuan-style braised beef were affected by the moisture content, oxidation of proteins and lipids, and integrity of the muscle fibers. Considering texture traits, when Sichuan-style pre-braised beef bought by consumers is stewed with other ingredients for about 30 min, its corresponding stewing time is 60 min in industrialized production processes. This process parameter can not only save energy consumption for practical production, but also improve the hardness value of the as-obtained Sichuan-style pre-braised beef, which is conducive to transportation through refraining from cracking of pre-braised beef pieces. When consumers only use simple heating to eat the Sichuan-style pre-braised beef product, stewing times of 120 or 150 min can be considered in industrialized production processes. This work provided a theoretical reference for the industrialized and standardized production of different types of prepared Sichuan-style braised beef.

## 1. Introduction

As a traditional dish, the Sichuan-style braised beef is easy to make and endows a spicy taste, which is loved by consumers. With the improvement of the economic level and the acceleration of life pace, the demand for prepared Sichuan-style braised beef continues to increase. Nevertheless, given the differences in production style, the physicochemical changes of beef during preparation are also quite different [1]. In China, due to the low threshold of the prepared meat industry, the relevant scientific research and unified production standards are severely lacking, and the quality of Sichuan-style braised beef is uneven, suffering from the limited industrial production as well.

Among the quality properties of meat products, texture properties are not only the main determinants of meat quality, but one of the most important factors affecting consumer acceptance [2]. Li et al. [3] pointed out that braised beef with tender texture has a better edible quality, and braised beef prepared on an open fire has a soft texture, endowing a hardness value of 766.81 ± 209.31 N and a chewiness value of 493.94 ± 94.24 N. Modzelewska-Kapituła, Tkacz, and Nogalski [1] reported that better tenderness is one of the important characteristics for cooked beef with desirable sensory characteristics. Although some researchers have explored the effect of processing methods on beef quality properties [4,5], there is less attention paid to specific recommendations for guiding the industrial production of a certain dish. Prepared Sichuan braised beef can be divided into two types according to how consumers handle it after purchase. The first is the prepared beef that consumers can directly eat after simple heating (usually this kind of prepared beef has a lower hardness value). The second is the prepared beef that needs further processing according to consumers’ preferences. How long it will take to stew the two types of prepared Sichuan-style beef to obtain their required texture properties and the formation mechanism of their texture properties under different stewing times deserve further investigation.

The present study aimed to demonstrate the effect of changes in the physicochemical properties of prepared Sichuan-style braised beef on its quality properties, especially the texture properties, caused by different stewing times. The ultimate goal is to provide a theoretical basis for the industrialized and standardized production of different types of prepared Sichuan-style braised beef, and manufacturers can reasonably adjust the stewing time according to different types of products.

## 2. Materials and Methods

### 2.1. Materials

The beef belly used in the experiment was from a commercial slaughterhouse in Ganzi Prefecture, Sichuan Province. The slaughtered cattle were 4-year-old female Tibet yaks, and the stunning method was transient high voltage current. After being stunned, the yaks were bled and then suspended on hooks. Then the yaks were skinned and eviscerated. Subsequently, the split yak meat was suspended in a 4 °C cold storage room for 72 h to discharge acid. Next, the beef belly was cut from the yak carcass, successively placed in a cooler (80 L, Ningbo Three Ants Outdoor Products Co., Ltd., Ningbo, China, approximately 6 °C), and transported it to the laboratory within 4.5 h. The pH value of fresh beef belly was 6.20 ± 0.03, while the pH value of beef belly delivered to the laboratory was 5.83 ± 0.03. The beef belly used in the experiment came from three yaks, and the sample of beef belly cut from each yak was about 6 kg. The whole experiments were repeated three times.

### 2.2. Sample Handling

The beef belly was cut into 3 cm^3^ pieces and divided into 6 equal portions for later use. Then rapeseed oil (20%), ginger (2%), green onions (2%), broad bean paste (15%), soy sauce (5%), oyster sauce (3%), cinnamon (1%), star anise (1%), fragrant leaves (0.25%), Chinese peppercorns (0.5%), fennel (0.5%), white sugar (2.5%), pepper (0.5%), chicken essence (0.5%) and chili (2.5%) were added to a pot (the added amount of each auxiliary ingredient was expressed as a percentage of meat weight). Then the added spices were fried for 2 min, and then water (the added water amount with the weight of five times the beef) and beef were added. After the water boiled, the different experimental groups were kept in the slightly boiling states (the temperature of water is maintained at about 95 °C) for 0, 30, 60, 90, 120, and 150 min, respectively. Finally, the beef belly was cooled at 4 °C and stored at 4 °C, and then random samples were analyzed for their physicochemical properties. All indicators need to be measured within two days.

### 2.3. Texture Properties

Refer to the method reported by Isleroglu, Kemerli, and Kaymak-Ertekin [6] to measure texture properties. The texture of the beef was determined with a texture analyzer (TA-XTPLUS, Stable Micro Systems, London, UK, the probe model is P/36R) at room temperature. The specific measurement parameters were as follows: test speed and return speed were 1.00 mm/s and 2.00 mm/s for two cycles, respectively, and trigger force was 5 g with target deformation of 30%.

### 2.4. Moisture Content and Moisture State

Moisture content was measured in accordance with the method reported by Alirezalu, Pirouzi, Yaghoubi, Karimi-Dehkordi, Jafarzadeh, and Mousavi Khaneghah [7]. About 5 g of ground beef samples was weighed in a dry glass plate and dried at 105 °C in a drier (BPG 9070A, Beijing Ever Bright Medical Treatment Instrument Co., Ltd., Beijing, China) until the difference between two weighing times was less than 2 mg. Finally, the moisture content was calculated according to the moisture loss of braised beef. Moisture state was determined in accordance with the method of Ishiwatari, Fukuoka, and Sakai [8] with some modifications. The moisture state of different treated braised beef samples (the size of the beef samples was about 1.5 cm × 1.5 cm × 1.5 cm) was examined by a low field nuclear magnetic resonance (LF-NMR) analyzer (MesoMR23-040V-I, Suzhou Niumag Analytical Instrument Corporation, Suzhou, China). The magnetic probe was a MesoMR23-040V-I, and the sequence choice was Q-CPMG. The remaining NMR parameters were as follows: SW = 100 KHz, SF = 20 MHz, RFD = 0.080 ms, RG1 = 20.0 db, P1 = 10.48 µs, DRG1 = 3, PRG = 2, TW = 2000 ms, NS = 4, p2 = 20.48 µs, TE = 0.800 ms, and NECH = 1500.

### 2.5. Observation of Microstructure

Samples for scanning electron microscopy (SEM) were prepared by referring to the method of Bolumar, Bindrich, Toepfl, Toldrá, and Heinz [9], and the micro-morphology was characterized using an Apreo 2C (Thermo Fisher Scientific, Franklin, MA, USA). The samples were affixed to the conductive adhesive, and then gold layers were sprayed onto the surface of the sample. The samples were observed at a magnification of 500 times, the accelerating voltage was 15.0 kV, and electron microscope pictures were obtained. Braised beef samples were cut into 1 × 0.5 × 0.5 cm^3^ pieces and soaked in PBS (pH = 7.2) containing 2.5% glutaraldehyde for 24 h. The beef sample was taken out and cleaned three times with PBS without glutaraldehyde. After cleaning, ethanol solutions with concentrations of 30%, 50%, 70%, and 90% were used for dehydration, respectively, and each concentration was treated for 1 h. The beef samples were freeze-dried in the instrument for 36 h (FDU-1200, EYELA, Tokyo, Japan), and the microstructure was observed by SEM.

### 2.6. Sarcoplasmic Protein (SP) and Myofibrillar Protein (MP)

SP and MP were extracted from beef belly based on the method of Suwandy, Carne, van de Ven, Bekhit, and Hopkins [10]. Furthermore, 25 g of ground beef sample was mixed with 100 mL of PBS (0.01 mol/L, containing 0.1 mol/L NaCl, 0.001 mol/L EGTA, and 0.002 mol/L MgCl_2_; pH = 7.0). After homogenization for 1 min at 8000 rpm with a homogenizer (FSH-2A, Shanghai Jipad Instrument Co., Ltd., Shanghai, China), the mixture was centrifuged (LC-LX-HLR210D, LICHEN, Changsha, China) at 7680× *g* for 15 min (4 °C), and the supernatant was primarily SP. The above operation was repeated three times. After adding 100 mL of PBS (0.01 mol/L, 0.1 mol/L NaCl) to the precipitate and homogenizing for 30 s (8000 rpm), the samples were centrifuged at 7680× *g* for 15 min (4 °C). The above operation was repeated two times. Finally, 100 mL of PBS (0.02 mol/L, pH = 6.0) was added to the precipitate, filtration was conducted after homogenization, and the filtrate was centrifuged (7680× *g*, 15 min, 4 °C). The remaining precipitate was MP.

### 2.7. Carbonyl and Sulfhydryl Content

The carbonyl and sulfhydryl contents were measured based on the method introduced by Guyon, Le Vessel, Meynier, and de Lamballerie [11]. The extracted protein was diluted to 5 mg/mL. Two protein solutions (0.8 mL) were taken from each group of samples and placed in a 5 mL centrifuge tube. One portion was treated with 1.6 mL of 3 mol/L HCl containing 0.3 wt.% 2,4-dinitrophenylhydrazine and was denoted as the treatment group. The other portion was treated with 1.6 mL of 3 mol/L HCl and was denoted as the blank group. After the treated and blank samples were reacted at room temperature for 30 min, 0.8 mL of 40 wt.% trichloroacetic acid was added to precipitate the proteins. After centrifuging at 4800× *g* for 5 min (4 °C) and removing the supernatant, 2 mL of mixed solution (ethanol:ethyl acetate = 1:1) was added to wash the precipitate and remove the unreacted 2,4-dinitrophenylhydrazine. After centrifuging again at 4800× *g* for 5 min (4 °C) repeated for three times, 3 mL of PBS (0.02 mol/L; pH 6.5; containing 6 mol/L guanidine hydrochloride) was added to dissolve the precipitate. After standing overnight, the absorbance was measured at 370 nm by an ultraviolet visible (UV) spectrophotometer (1900, Suzhou xiaotianyuan Instrument Equipment Co., Ltd., Suzhou, China), and the carbonyl content of protein was calculated according to the molar extinction coefficient (22,000/M·cm).

For the determination of sulfhydryl content, 1 mL of protein solution was mixed with 9 mL of PBS (0.05 mol/L; pH = 7; containing 8 mol/L urea, 0.6 mol/L NaCl, and 0.01 mol/L EGTA). Then, 3 mL of diluted solution was mixed with 0.4 mL of 0.1% 2-nitrobenzoic acid. Next, it was placed in darkness at 40 °C for 25 min. After cooling to room temperature, the absorbance was measured at 412 nm by an UV spectrophotometer (1900, Suzhou xiaotianyuan Instrument Equipment Co., Ltd., Suzhou, China). Sulfhydryl content was calculated according to the molar extinction coefficient (13,600/M·cm).

### 2.8. Endogenous Fluorescence Intensity

The endogenous fluorescence intensity of beef protein was determined based on the method proposed by Zhang, Li, Diao, Kong, and Xia [12] with slight modifications. MP was diluted to 0.5 mg/mL with PBS (0.02 mol/L; pH = 6.5), and the spectra at 300–400 nm were recorded with a fluorescence spectrophotometer (FLUOROMAX-4cp, HORIBA Jobin Yvon, Piscataway, NJ, USA). The excitation wavelength and the scanning speed were 295 nm and 1500 nm/min, respectively, and the excitation voltage was 400 V.

### 2.9. Thiobarbituric Acid Reactive Substances (TBARS)—Expressed as mg of Malondialdehyde (MDA) per kg of Beef

The TBARS value in beef was determined in accordance with the method of Domínguez, Gómez, Fonseca, and Lorenzo [13] with some modifications. Furthermore, 10 g of ground beef samples was added with 20 mL of trichloroacetic acid solution (20%, *v*/*v*) and homogenized for 1 min at 5000 rpm. After centrifuging at 4800× *g* for 10 min (4 °C), 5 mL of the supernatant was collected into a colorimetric tube, and the same volume of 2-thiobarbituric acid solution (0.02 mol/L) was added. The mixture was heated at 95 °C for 20 min in darkness. Absorbance was measured at 532 nm after cooling to room temperature. The TBARS value was expressed as mg of malondialdehyde (MDA) per kg of beef.

### 2.10. Fourier Transform Infrared Spectroscopy (FT-IR)

FT-IR spectra of MP were obtained in accordance with the method of Wang, Kong, Li, Liu, Zhang, and Xia [14] with some modifications. The extracted beef MP is freeze dried and then ground into a powder. Potassium bromide was mixed with MP powder in a ratio of 100:1, and the mixture was ground in an agate mortar. A small amount of ground powder was put into a tablet press for tablet pressing and then measured by FT-IR spectrometry (Spectrum Two N, PerkinElmer, Shelton, CT, USA). The scanning conditions were as follows: range of 4000–400 cm^-1^, resolution of 4 cm^-1^, and frequency of 64.

### 2.11. Statistical Analysis

Statistical analyses were performed using SPSS, ANOVA on moisture content, carbonyl contents, sulfhydryl content, and TBARS were performed using Tuckey post hoc tests as appropriate. The level of significance was set at *p* < 0.05. Hierarchical cluster analysis (HCA) was performed using R software version 3.6.3 (heat map package) to explore the relationship between variables. The experiments were repeated 3 times in a row on the same day while at different points in time.

## 3. Results and Discussion

### 3.1. Texture Properties

Meat texture properties is an important characteristic in terms of consumer preference and satisfaction [15]. Texture properties of Sichuan-style braised beef stewed at different times are shown in Table 1. When stewing for 60 min, the hardness value of Sichuan-style braised beef increased from 203.94 g to 2227.07 g, and its chewiness value increased from 81.50 g to 756.00 g, thereby the differences were remarkable (*p* < 0.05). With further increased stewing time (>60 min), the values of hardness and chewiness of Sichuan-style braised beef obviously decreased (*p* < 0.05). When the stewing time reached 150 min, the values of hardness and chewiness were 1065.95 and 327.27 g, respectively. Nevertheless, with increased stewing time, the value of springiness and cohesiveness decreased from 0.77 to 0.67 and 0.55 to 0.44, respectively.

The hardness, chewiness, and springiness can affect consumer’s acceptance, especially for Chinese people who prefer meat products with low hardness and chewiness, as well as good springiness [16]. The different texture properties of braised beef may be due to the changes in the physicochemical properties of beef protein and fat caused by different stewing times, resulting in differences of microstructure, as well as the changes in moisture content and state, eventually leading to different texture properties. Among them, the changes of MP and muscle fibers are the main factors affecting the structural characteristics of beef. In a short stewing time (<60 min), the MP of beef undergoes thermal denaturation, and thermal contraction occurs on the muscle fiber, resulting in increased hardness and chewiness. With the further extension of the stewing time (>60 min), MP, collagen, and connective tissue could be degraded to a certain extent, and the muscle fiber is broken, thereby causing a downward trend in hardness, springiness, cohesiveness, and chewiness. Gagaoua et al. [17] pointed out that the composition and content of protein were different in beef samples with different tenderness. This also indicates that differences in protein can have an impact on the texture properties of beef.

### 3.2. Moisture Content and Moisture State

The moisture content of meat products has an important influence on meat quality and tenderness [12]. As illustrated in Table 2, with prolonged stewing time (0–90 min), the moisture content of Sichuan-style braised beef significantly decreased (*p* < 0.05) (from 70.268% to 59.762%). When the stewing time increased to over 90 min, the moisture content had no marked change (*p* > 0.05). When the stewing time was controlled from 0 to 90 min, the moisture content in the beef obviously decreased. This finding was due to the moisture in the muscle fiber gaps being squeezed out upon the thermal contraction of the beef, resulting in significantly reduced moisture content [5]. The oxidation of beef protein and fat during heating could also lead to a decrease in the binding capacity of beef protein to moisture, which in turn suffers from decreased moisture content. Elmas et al. [18] pointed out that a higher moisture content and moisture activity of meat products corresponded with better springiness. The change of moisture content of braised beef was similar to that of springiness (Table 1), further confirming Elmas’ opinion.

Magnetic resonance imaging (MRI) is a nondestructive method that has been widely used in studying water distribution among meats, and the brightness of the meat sample image can reflect its moisture content [19]. As shown in Figure 1a, the brightness of the MRI images significantly decreased with prolonged stewing time. NMR results showed that the moisture content of beef decreased significantly with prolonged stewing time. This result was consistent with the findings in Table 2. In the transverse relaxation time measured by LF-NMR, T_21_, T_22_, and T_23_ represent the bound moisture, the immobilized moisture, and the free moisture, respectively [20]. The transverse relaxation time and peak value of braised beef samples changed with prolonged stewing time, indicating that prolonged stewing time promoted the mutual transformation of moisture state in beef samples. As shown in Table 3, T_21_ has a small change in peak ratio, whereas T_22_ and T_23_ have a large change in peak ratio. With prolonged stewing time, the peak proportion of T_22_ decreased (from 94.84% to 81.08%), whereas that of T_23_ increased (from 4.09% to 18.40%). This finding indicated that with prolonged stewing time, the tissue structure of beef was damage and the microstructure also changed, resulting in decreased binding force of meat to moisture, and part of the immobilized moisture being transformed into free moisture. Cheng et al. [21] found that a lower content of immobilized moisture in meat products corresponded with poorer springiness. The change of immobilized moisture content of braised beef in this study was similar to that of springiness (Table 1), which further confirmed Cheng’s opinion.

### 3.3. SEM Analysis

To elucidate the effect of stewing on beef structure, SEM characterization was performed to study the changes of beef microstructure. As Figure 2 demonstrates, the longitudinal section of the muscle fibers of fresh beef were neatly arranged, and many gaps existed between the muscle fibers, and the sum of the diameter of the three muscle fibers was slightly smaller than 200 μm. Some researchers have pointed out that moisture can be retained between the muscle gaps [22,23]. With prolonged stewing time (0–60 min), due to thermal contraction, the gap size between muscle bundles decreased, and many muscle bundles combined together, making the structure of beef more compact, and this phenomenon became particularly obvious when stewing time reached up to 60 min, and the sum of the diameter of the four muscle fibers was about 200 μm. This indicated that the moisture in the beef was squeezed out and the tissue structure became denser, which was well consistent with the increased hardness (Table 1) and the decreased moisture content of beefs (Table 2).

With further prolonged stewing time (60–150 min), the boundaries between the muscle bundles became blurred. Similar results were also obtained from the cross-section SEM image. When stewing time exceeded 60 min, some muscle fibers were broken, which can be speculated that the prolonged stewing time causes the destruction and oxidation degradation of muscle protein, and constitutionally leads to a loss of toughness of the muscle fibers that make up the beef tissue. When stewing time exceeded 60 min, the diameter of muscle fibers appeared larger, and the total diameter of three muscle fibers is slightly greater than 200 μm, as a consequence of the damage of muscle fiber membrane, accompanied by muscle fibers being looser. This result is consistent with previously reported works. For instance, Zhao et al. [24] also pointed out that the diameter of muscle fiber significantly increased in the braised meat. Compared with fresh beef muscle fiber, fewer connective tissue or muscle fiber membranes can be observed on the microstructure of beef muscle fiber after stewing for 120 or 150 min. This is mainly due to the collagen of connective tissue or muscle fiber membrane becoming gelatinized and having flowed out during the long-term stewing process, suffering from the destruction of the muscle fiber integrity, and finally resulting in the softened beef tissue and the corresponding changes in the microstructure. The changes in microstructures were consistent with the changes in texture properties (Table 1) and moisture (Table 2).

### 3.4. Protein Oxidation

As shown in Table 2, the carbonyl contents of MP and SP in braised beef increased significantly (*p* < 0.05) with prolonged stewing time. When the stewing time reached 150 min, the carbonyl content of SP increased from 0.04 to 2.44 nmol/mg, and the carbonyl content of MP increased from 0.60 to 5.79 nmol/mg. Choi and Chin [25] pointed out that the cooking process leads to the denaturation and oxidation of proteins in meat products, resulting in increased carbonyl content and protein cross-linked polymers. Gatellier, Kondjoyan, Portanguen, and Santé-Lhoutellier [26] also found that the extension of cooking time and the increase in cooking temperature had significant effects on the carbonyl content of beef. Utrera, Morcuende, and Estévez [27] further pointed out that high temperatures promoted the occurrence of oxidative reactions during cooking, resulting in the formation of carbonyl derivatives. In addition, the protein peptide skeleton through α-amidation and covalent binding to non-protein carbonyl compounds can also lead to increased protein carbonyl content [28]. In China, there is no clear regulation on carbonyl content in braised beef. Estévez [29] pointed out that carbonyl content in cooked meat products was about 5 nmol/mg protein, which was consistent with the results of this study.

As the most active functional group in protein, the reduction of sulfhydryl content is often used as an indicator of protein oxidation [30]. As shown in Table 2, the total sulfhydryl contents of SP and MP decreased significantly with prolonged stewing time (*p* < 0.05). When the stewing time reached 150 min, the sulfhydryl content of SP decreased from 165.20 to 8.82 nmol/mg, and the sulfhydryl content of MP decreased from 68.10 to 40.89 nmol/mg. The sulfhydryl groups in beef are prone to oxidation reactions during stewing [31], and cysteine reacts with free-radical species to generate sulfur center free radicals, which can further react with the sulfhydryl or mercaptan salts of another protein molecule to form protein disulfide bonds, resulting in reduced sulfhydryl content.

### 3.5. Endogenous Fluorescence Intensity

Tryptophan residues are located at the core of the natural protein structure and have high fluorescence intensity, they are easily attacked by reactive oxygen species and undergo oxidation reactions, resulting in decreased protein fluorescence intensity [32]. Therefore, the loss of tryptophan fluorescence can be used as one of the indicators of protein-oxidation degree. As shown in Figure 3, with prolonged stewing time, the endogenous fluorescence intensity of SP and MP in beef significantly decreased. Results showed that the prolonged stewing time increased the oxidation of beef protein. Mitra, Lametsch, Akcan, and Ruiz-Carrascal [33] also found that high temperature and long stewing time can significantly reduce the endogenous fluorescence intensity of meat protein. The decrease in endogenous fluorescence intensity of beef protein confirmed that beef protein also underwent oxidation during stewing, consistent with the results of increased carbonyl content and decreased sulfhydryl content in Table 2.

### 3.6. TBARS

Lipids are one of the chemically unstable food components, which readily undergo oxidation reactions and are closely related to the oxidation of proteins [27]. The TBARS value can be used to evaluate the oxidation degree of lipids in meat and meat products. With prolonged stewing time, the TBARS values of beef belly was examined, and results are shown in Table 2. The TBARS value of beef increased from 0.02 to 0.05, 0.06, 0.11, 0.11, and 0.12 mg/kg when the beef was stewed for 30, 60, 90, 120, and 150 min, respectively. In China, there is no clear regulation on TBARS value for braised beef. Zhou et al. [34] proposed that the content of TBARS in meat products usually ranged from 0.1 to 10 mg/kg. In this work, when beef belly was stewed for 150 min, the TBARS value was 0.12 mg/kg, which was relatively low and generally had no effect on human health. With prolonged stewing time, the TBARS value of beef increased overall, indicating that the extension of stewing time promoted the oxidation of beef lipids to a certain extent. Some researchers have pointed out that the free radicals and MDA generated by lipid oxidation in the food system can also cause protein oxidation [20]. The TBARS results were consistent with the increase in beef protein carbonyl content (Table 2), the decrease in sulfhydryl content (Table 2), and the change in protein structure (Figure 4).

### 3.7. Protein Secondary Structure

As shown in Figure 4, the secondary structure of MP in beef changed significantly with prolonged stewing time. When the stewing time was 0 min, α-helix was the most dominant secondary structure (37.40%) in beef MP. With prolonged stewing time, the α-helix gradually decreased from 37.40% (0 min) to 32.36% (150 min) (*p* < 0.05), and β-sheet increased gradually from 15.54% (0 min) to 29.15% (150 min) (*p* < 0.05). The β-turn angle gradually decreased from 24.24% to 20.21%, and the random coil gradually decreased from 22.81% to 18.27%. During heating, the structure of MP expanded [35], resulting in the reduction of intramolecular hydrogen bonds and the instability of the secondary structure of MP. Finally, a part of the α-helix structure in the MP molecule was converted into β-sheet. Vaskoska et al. [36] also showed that heating led to a reduction in the α-helix structure of meat proteins. Zhang et al. [37] pointed out that oxidation leads to changes in the secondary structure of proteins, resulting in the mutual transformation of α-helix, β-sheet, β-turn, and random coil. The changes in β-turn and random coil were relatively small, which may be due to the mutual transformation between β-turn, random coil, or other loose structures caused by the changes in intermolecular and intramolecular chemical forces during stewing. Consequently, their total changes were relatively small.

### 3.8. HCA of Sichuan-Style Braised Beef Treated by Different Stewing Times

As shown in Figure 5, HCA divides six kinds of stewing time into four categories. Clusters X (X = 1, 2, 3, and 4) correspond to the stewing times of 0, 30/60, 90, and 120/150 min, respectively. The difference between cluster 2 and cluster 3 is obvious, indicating that the physicochemical properties of braised beef change significantly when the stewing time exceeded 60 min. This is consistent with the results of the changes of each indicator of braised beef. HCA divides ten indicators into three categories. Cluster 1 includes MP total sulfhydryl content, springiness, moisture content, cohesiveness, and SP carbonyl content. The overall indicators in cluster 1 are negatively correlated with the stewing time. The longer the stewing time is, the smaller the values of the abovementioned indicators are. Cluster 2 includes hardness and chewiness. The values of these two indicators first increase and then decrease with prolonged stewing time. Cluster 3 includes MP carbonyl content, SP carbonyl content, and TBARS. The indicators in cluster 3 are positively correlated with the stewing time. The longer the stewing time is, the bigger the values of these indicators are. In addition, TBARS is positively correlated with MP carbonyl content and SP carbonyl content, indicating that lipid oxidation in braised beef is accompanied by protein oxidation during the stewing process. The results show that the physicochemical properties of braised beef change greatly with prolonged stewing time. Therefore, based on the texture properties, when Sichuan-style pre-braised beef bought by consumers is stewed with other ingredients for 30 min, its corresponding stewing time is 60 min in industrialized production processes. This process parameter can not only save energy consumption for practical production, but can also improve the hardness value of the as-obtained Sichuan-style pre-braised beef, which is conducive to transportation through refraining from crack of pre-braised beef pieces. When consumers only apply simple heating to eat Sichuan-style pre-braised beef products, stewing times of 120 or 150 min can be considered in industrialized production processes.

## 4. Conclusions

During the stewing process of Sichuan-style braised beef, prolonged stewing time promoted the oxidation degree of beef protein and lipid. With prolonged stewing time, the muscle fibers of beef initially contracted and then broke. The formation mechanism of the texture properties of braised beef can be summarized as the thermal contraction of beef muscle fibers in a short stewing time (<60 min), which reduced the moisture content in the beef. Moreover, the protein oxidation produced cross-linking, ultimately increasing the hardness value of the beef. Nevertheless, with further prolonged stewing time (>60 min), the protein in the beef underwent obvious oxidation degradation, ultimately reducing the hardness value of the beef. Therefore, considering the texture properties, when Sichuan-style pre-braised beef bought by consumers is stewed with other ingredients for about 30 min, its corresponding stewing time is 60 min in industrialized production processes. When consumers only apply simple heating to eat Sichuan-style pre-braised beef products, stewing time for 120 or 150 min can be considered in industrialized production processes. Notably, according to the research results in this work, we will gradually scale up the experiment in the cooperative meat processing enterprises in future works, and finally realize the industrialized production of Sichuan-style braised beef, so that people all over the world can eat authentic Sichuan-style braised beef.

## Figures and Tables

**Figure 1 foods-12-01386-f001:**
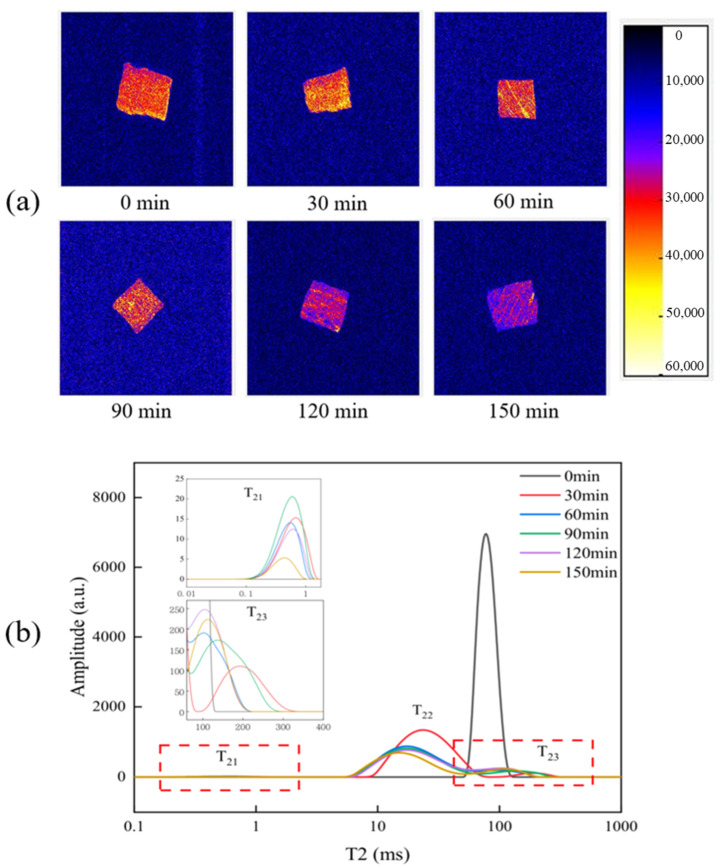
Effects of stewing time on the moisture distribution (**a**) and transverse relaxation time T_2_ (**b**) of Sichuan-style braised beef.

**Figure 2 foods-12-01386-f002:**
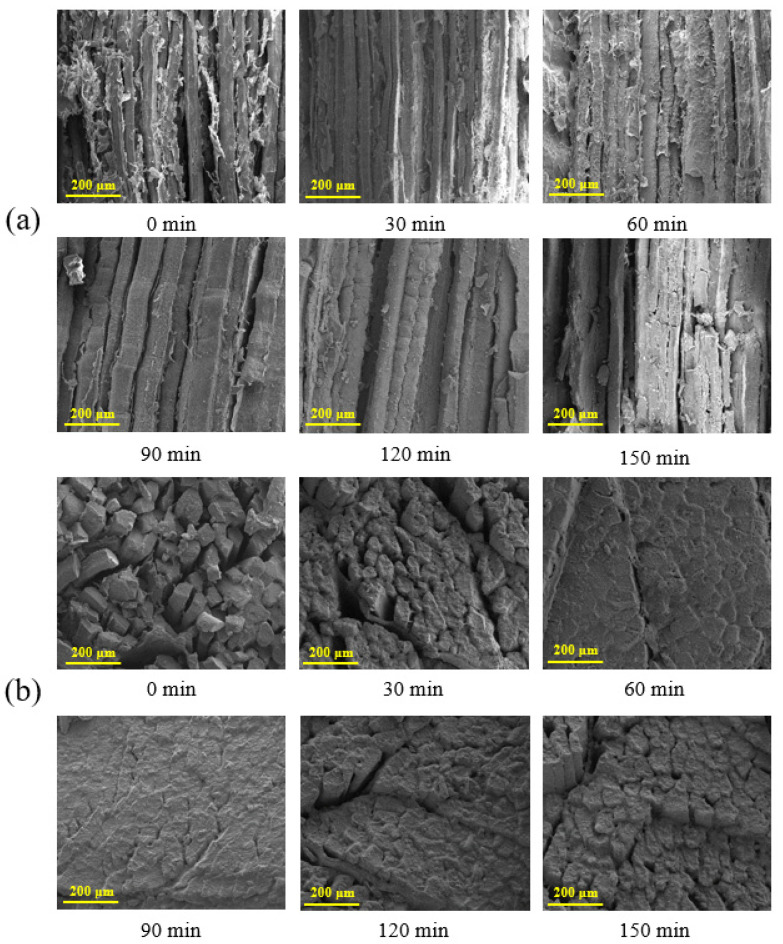
Analysis of Sichuan-style braised beef microstructure by SEM. (**a**) Longitudinal section, (**b**) cross-section.

**Figure 3 foods-12-01386-f003:**
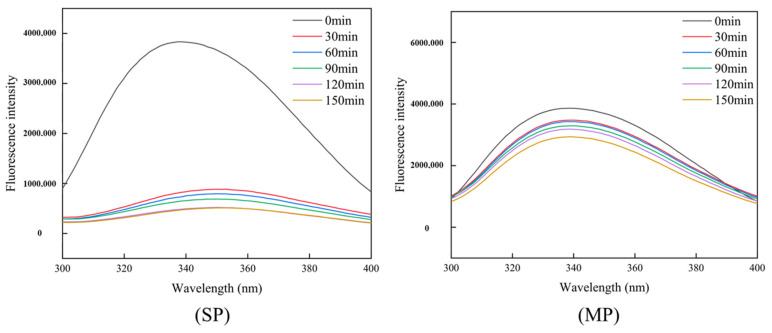
Effects of stewing time on endogenous fluorescence intensity of Sichuan-style braised beef protein.

**Figure 4 foods-12-01386-f004:**
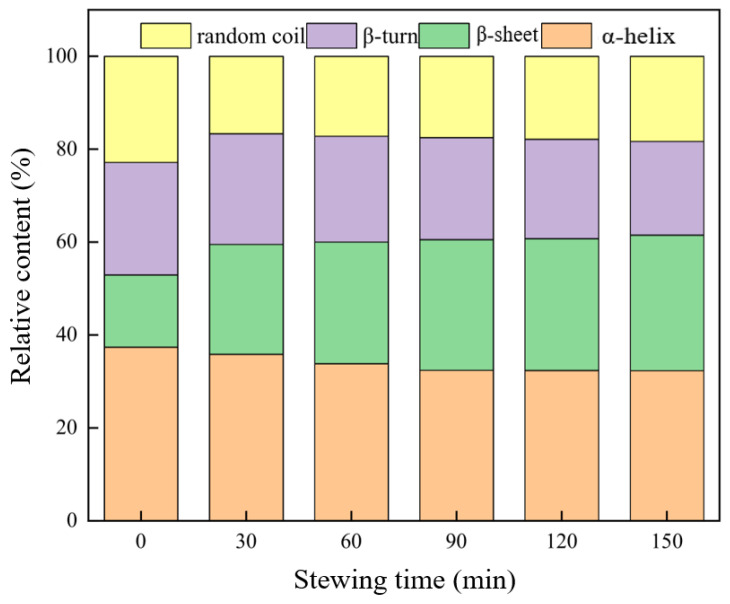
Effects of stewing time on the secondary structure of MP in Sichuan-style braised beef.

**Figure 5 foods-12-01386-f005:**
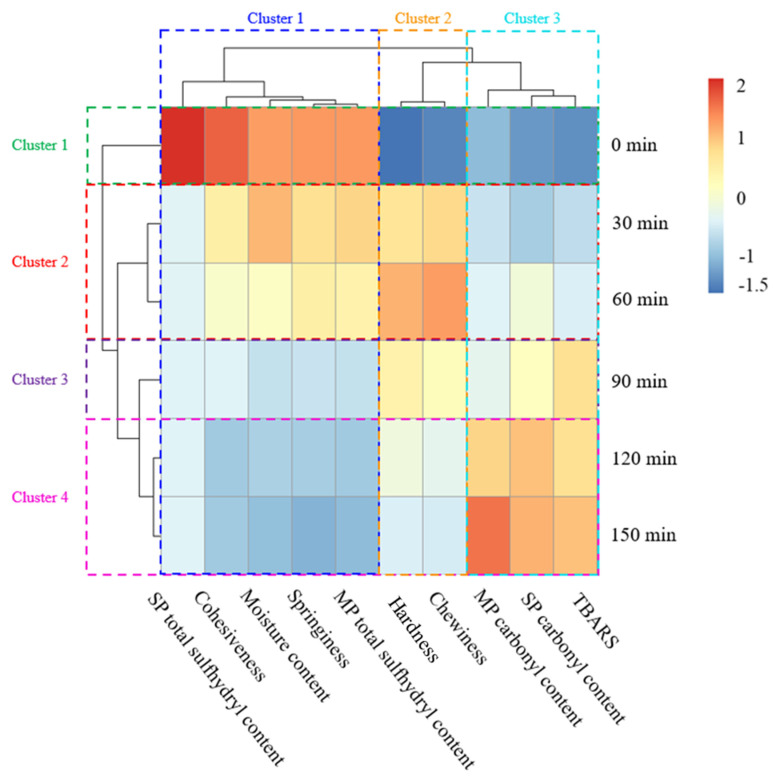
HCA of Sichuan-style braised beef treated by different stewing times.

**Table 1 foods-12-01386-t001:** Effects of stewing time on texture properties of Sichuan-style braised beef.

Stewing Time (min)	Hardness (g)	Springiness (Ratio)	Cohesiveness (Ratio)	Chewiness (g)
0	203.94 ± 21.02 ^a^	0.77 ± 0.01 ^a^	0.55 ± 0.01 ^a^	81.50 ± 4.08 ^a^
30	1896.23 ± 478.87 ^bc^	0.75 ± 0.02 ^a^	0.50 ± 0.04 ^a^	651.34 ± 132.05 ^cd^
60	2227.07 ± 87.96 ^c^	0.73 ± 0.09 ^a^	0.48 ± 0.03 ^a^	756.00 ± 100.15 ^d^
90	1721.12 ± 81.06 ^bc^	0.69 ± 0.06 ^a^	0.46 ± 0.10 ^a^	507.64 ± 45.81 ^bc^
120	1298.48 ± 439.08 ^b^	0.68 ± 0.08 ^a^	0.45 ± 0.03 ^a^	372.38 ± 52.75 ^b^
150	1065.95 ± 47.77 ^b^	0.67 ± 0.07 ^a^	0.44 ± 0.03 ^a^	327.27 ± 17.87 ^ab^

Values represent the average ± SEM, SEM stands for standard error of mean. The superscripts (a–d) of different lowercase letters in the same column indicate significant differences (*p* < 0.05).

**Table 2 foods-12-01386-t002:** Effects of stewing time on physiochemical properties of Sichuan-style braised beef.

	Stewing Time (min)					
	0	30	60	90	120	150
Moisture content (%)	70.27 ± 1.03 ^c^	69.19 ± 1.23 ^c^	63.93 ± 0.52 ^b^	59.76 ± 0.01 ^a^	58.89 ± 0.15 ^a^	57.81 ± 0.99 ^a^
MP carbonyl content (nmol/mg)	0.60 ± 0.01 ^a^	1.45 ± 0.004 ^b^	1.84 ± 0.010 ^c^	2.04 ± 0.01 ^d^	4.40 ± 0.004 ^e^	5.79 ± 0.01 ^f^
SP carbonyl content (nmol/mg)	0.04 ± 0.00 ^a^	0.51 ± 0.01 ^b^	1.27 ± 0.01 ^c^	1.50 ± 0.004 ^d^	2.31 ± 0.05 ^e^	2.44 ± 0.02 ^f^
MP total sulfhydryl content (nmol/mg)	68.10 ± 2.88 ^c^	63.11 ± 4.08 ^bc^	57.84 ± 1.85 ^b^	45.83 ± 0.87 ^a^	42.93 ± 0.71 ^a^	40.89 ± 1.88 ^a^
SP total sulfhydryl content (nmol/mg)	165.20 ± 2.55 ^c^	12.99 ± 0.17 ^b^	11.030 ± 0.30 ^ab^	10.54 ± 0.92 ^ab^	10.21 ± 0.32 ^ab^	8.82 ± 0.30 ^a^
TBARS (mg/kg)	0.02 ± 0.001 ^a^	0.05 ± 0.001 ^b^	0.06 ± 0.002 ^c^	0.11 ± 0.01 ^d^	0.11 ± 0.001 ^d^	0.12 ± 0.001 ^d^

Values represent the average ± SEM, SEM stands for standard error of mean. The superscripts (a–f) of different lowercase letters in the same line indicate significant differences (*p* < 0.05).

**Table 3 foods-12-01386-t003:** Effects of stewing time on the proportion of LF-NMR peaks of Sichuan-style braised beef.

Stewing Time (min)	T_21_ (%)	T_22_ (%)	T_23_ (%)
30	1.08	94.84	4.09
60	1.12	87.64	11.24
90	1.86	85.18	12.96
120	1.08	81.68	17.24
150	0.52	81.08	18.40

## Data Availability

Data is contained within the article.

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
