# Peer review of "Changes in Texture Characteristics and Special Requirements of Sichuan-Style Braised Beef for Industrial Production: Based on the Changes in Protein and Lipid of Beef"

_foods, 2023, doi:10.3390/foods12071386_

Round 1
Reviewer 1 Report
please give brief description about the methodologies used for measuring the parameters on sections from 2.3 to 2.11
please follow the journal instruction for references style
Author Response
Q1: Please give brief description about the methodologies used for measuring the parameters on sections from 2.3 to 2.11.
Response: We thank the reviewer for this valuable suggestion, we are very sorry for the incomplete description in the original manuscript. We have added the relevant description about the methodologies used for measuring the parameters on sections from 2.3 to 2.11 in the revised manuscript.
Q2: Please follow the journal instruction for references style.
Response: We are very appreciative of the reviewer for the very professional comment. According to your constructive suggestion, we have checked out references style following the journal instruction thoroughly and made proofreading accordingly.
Reviewer 2 Report
The Manuscript is about the assessment of possibility to standardizing the production of Sichuan-style braised beef. Currently, industrial production of this meal in China is limited.
The Authors found it important to assess changes in sensory properties during cooking to find appropriate cooking time for industrial production of a traditional meal.
Yak meat as an object of research is very interesting.
The Manuscript is clear, relevant for the field and presented in a well-structured manner.
Some more information is recommended to add or correct
2.1. Materials:
a) what was the type of slaughter – electrical or gas stunning, other? Please, describe briefly the carcass processing stages until belly was cut out of the carcass.
b) was the pH (5.58) at the arrival to the lab or at slaughter? Please, provide both pH values.
c) were the bellies transported in a cooler wrapped?
d) how many female Tibet yaks were slaughtered for each experiment?
e) how long was a period between slaughter and delivery to the lab?
2.2. Sample handling
a) How long and under what temperature the samples were stored before further use?
b) What was the final cooked beef temperature after cooling?
c) please, describe soup, which was used as an object for SDS-PAGE (see 3.8.)
3.2. Moisture content and moisture state
Please correct “ The brightness of the magnetic resonance imaging image …..”
3.3. SEM analysis
Figure 2. Please add magnifications.
To confirm changes in the beef structure depending on stewing time, quantitative results of microstructural analysis, ex., the size of the muscle bundle / diameter of myofibrils, description of the connective tissue, etc. are recommended to provide.
- when the Authors describe protein and lipid oxidation during stewing the safety aspects should be discussed as well: can the products of oxidation in described amounts affect consumer safety during regular consumption of the Sichuan-style braised beef prepared as recommended in the Manuscript.
3.8. SDS-PAGE
- please explain how the “soup” was obtained
- Figure 5 seems to be very uninformative. PAGE patterns carry little information. It is quite obvious that during the cooking process, denaturation and aggregation of proteins occur. And proteins stop at the top of a gel. The applied technique appears not to be quite applicable to obtaining a visual picture of protein changes during stewing.
The picture shown in Fig. 5c indicates an increase part of hydrolyzed proteins that form the "darkening" of gels 120 and, especially, after 150 min of thermal treatment. These patterns also are not informative.
p. 11. Please present more arguments to specify the choice of the optimal stewing time of 60 and 90 minutes. The choice is not evident because these regimes of thermal treatment result in maximal beef hardness and chewiness.
How long should further treatment be, over 60 min, to get a ready -to-eat Sichuan-style braised beef with required tenderness, chewiness and taste?
Author Response
The Manuscript is about the assessment of possibility to standardizing the production of Sichuan-style braised beef. Currently, industrial production of this meal in China is limited. The Authors found it important to assess changes in sensory properties during cooking to find appropriate cooking time for industrial production of a traditional meal. Yak meat as an object of research is very interesting. The Manuscript is clear, relevant for the field and presented in a well-structured manner. Some more information is recommended to add or correct.
Response: Thank you very much for your positive comments on our manuscript. Your recognition and suggestions have given us great encouragement. All requested revisions based on the professional comments from the reviewer have been carried out.
Specific comments:
Q1: 2.1. Materials:
- a) what was the type of slaughter – electrical or gas stunning, other? Please, describe briefly the carcass processing stages until belly was cut out of the carcass.
- b) was the pH (5.58) at the arrival to the lab or at slaughter? Please, provide both pH values.
- c) were the bellies transported in a cooler wrapped?
- d) how many female Tibet yaks were slaughtered for each experiment?
- e) how long was a period between slaughter and delivery to the lab?
Response: We thank the reviewer for the very professional questions and fully agree that more experimental parameters should be provided. According to reviewer’s opinion, we have supplemented relevant information in our revised manuscript. Now, we copy the relevant part here for your check:
“2.1. Materials
The beef belly used in the experiment was from a commercial slaughterhouse in Ganzi Prefecture, Sichuan Province. The slaughtered cattle were 4-year-old female Tibet yaks, and the stunned method was transient high voltage current. After being stun, the yaks were bled and then suspended on the hook. Then the yaks were skinned and eviscerated. Subsequently, the split yak meat was suspended in a 4 °C workshop for 72 h to discharge acid. Next, the beef belly was cut from the yak carcass, successively put it in a cooler (80 L, Ningbo Three Ants Outdoor Products Co., Ltd, China, approximately 6 °C) and transported it to the laboratory within 4.5 h. The pH value of fresh beef belly was 6.20±0.03, while the pH value of beef belly delivered to the laboratory was 5.83±0.03. The beef belly used in the experiment came from three yaks, and the beef belly cut from each yak was about 6 kg. The whole experiments were repeated three times.”
Q2: 2.2. Sample handling
- a) How long and under what temperature the samples were stored before further use?
- b) What was the final cooked beef temperature after cooling?
- c) please, describe soup, which was used as an object for SDS-PAGE (see 3.8.)
Response: Thank you very much for your very professional question, we deeply apologize that we didn't give complete information in the previous manuscript. All samples were stored at 4 ℃ and all indicators should be determined within two days. Considering the scientific nature and preciseness, we deleted the relevant discussion on the electrophoretic experiment in the revised manuscript.
Q3: 3.2. Moisture content and moisture state
Please correct “The brightness of the magnetic resonance imaging image …..”
Response: Thank you very much for your scientific advice, we have modified the sentence in the revised manuscript.
Q4: 3.3. SEM analysis
Figure 2. Please add magnifications.
To confirm changes in the beef structure depending on stewing time, quantitative results of microstructural analysis, ex., the size of the muscle bundle / diameter of myofibrils, description of the connective tissue, etc. are recommended to provide.
Response: Thank you very much for your kind reminder. We have added magnification in Figure 2. Additionally, we have supplemented relevant description about microstructural changes in the beef structure depending on stewing time in the revised manuscript. Now, we copy the relevant part here for your check:
“When stewing time exceeded 60 min, the diameter of muscle fibers appears larger, and the total diameter of three muscle fibers is slightly greater than 200 μm, as a consequence of the damage of muscle fiber membrane, accompanied by muscle fibers looser. This result is well consistent with previously reported works. For instance, Zhao et al. [25] also pointed out that the diameter of muscle fiber significantly increased in the braised meat. Com-pared with fresh beef muscle fiber, fewer connective tissue or muscle fiber membrane can be observed on the microstructure of beef muscle fiber stewing for 120 or 150 min. This is mainly due to the collagen of connective tissue or muscle fiber membrane gelatinized and flowed out during the long-term stewing process, suffering from the destruction of the muscle fiber integrity, and finally resulting in the softened beef tissue and the corresponding changes in the microstructure.”
Q5: When the Authors describe protein and lipid oxidation during stewing the safety aspects should be discussed as well: can the products of oxidation in described amounts affect consumer safety during regular consumption of the Sichuan-style braised beef prepared as recommended in the Manuscript.
Response: Thank you so much for this kind reminding. We have added the necessary information in section 3.4 and section 3.6 of the revised manuscript. Now, we copy the relevant part here for your check:
“In China, there is no clear regulation on carbonyl content in braised beef. Estévez [30] pointed out that carbonyl content in cooked meat products was about 5 nmol/mg protein, which was consistent with the results of this study; In China, there is no clear regulation on TBARS value for braised beef. Zhou et al. [35] proposed that the content of TBARS in meat products usually ranged from 0.1 to 10 mg/kg. In this work, when beef belly was stewed for 150 min, the TBARS value was 0.12 mg/kg, which was relatively low and generally have no effect on human health.”
Q6: 3.8. SDS-PAGE
- Please explain how the “soup” was obtained
- Figure 5 seems to be very uninformative. PAGE patterns carry little information. It is quite obvious that during the cooking process, denaturation and aggregation of proteins occur. And proteins stop at the top of a gel. The applied technique appears not to be quite applicable to obtaining a visual picture of protein changes during stewing.
The picture shown in Fig. 5c indicates an increase part of hydrolyzed proteins that form the "darkening" of gels 120 and, especially, after 150 min of thermal treatment. These patterns also are not informative.
Response: We appreciate the reviewer for very instructive comments. As the reviewer described, protein could denature and aggregate during the cooking process, which may lead to protein aggregation on the top of the gel, resulting in less obvious results. Although results of electrophoretic experiment give some changes, few information can be obtained. Considering the scientific nature and preciseness, we deleted the relevant discussion on the electrophoretic experiment in the revised manuscript.
Q7: p. 11. Please present more arguments to specify the choice of the optimal stewing time of 60 and 90 minutes. The choice is not evident because these regimes of thermal treatment result in maximal beef hardness and chewiness.
How long should further treatment be, over 60 min, to get a ready to eat Sichuan-style braised beef with required tenderness, chewiness and taste?
Response: We appreciate the reviewer for the very professional reminder and apologize for the ambiguity in our original description. According to your constructive comments, we have rewritten the relevant part in the section 3.8 of revised manuscript, and then copy the relevant content here for your check:
“Therefore, based on the texture properties, when Sichuan-style pre-braised beef bought by consumers is stewed with other ingredients for about 30 min, and the corresponding its stewing time is 60 min in industrialized production processes. This process parameter can not only save energy consumption for practical production, but also improve the hardness value of the as-obtained Sichuan-style pre-braised beef, which is conducive to transportation through refraining from crack of pre-braised beef pieces. When consumers only give simple heating to eat Sichuan-style pre-braised beef product, stewing time for 120 or 150 min can be considered in industrialized production processes.”
Reviewer 3 Report
Authors present an interesting aspect of texture quality for a certain local dish. They provide valuable information about its quality and how it should be prepared. There are some points that should be clarified in order the manuscript to provide more proper information and also to meet the standards of the journal offering also an adding value to the readers.
Please see my comments bellow:
Introduction--> Could you please refer which is the optimal texture for Sichuan-style braised beef according to preferences of consumers and if deviations from that texture are observed in the two products you refer?
2.1 the cattle slaughtered --> the slaughtered cattle
2.2. specify the mass (gr) of each portion. In addition the percentages where do they refer to? final volume? final mass?
2.3-2.12 kindly describe briefly the methods you use and the initial volume or mass of the ingredient you use. In addition specify how many replicated did you apply. In 2.12 specify also how you conduct cluster analysis
Table 1 specify the units if applicable in each trait. Please set also the superscripts a,b,c ... starting from the value with the lowest measure (the same in all tables)
Conclusions--> Authors suggest a stewing time 90 min or 60 min according to the type of product. Do the respective texture traits falls into consumers' expectations? Authors could make such a link even with previous reported results to justify stronger the results of their study.
Author Response
Authors present an interesting aspect of texture quality for a certain local dish. They provide valuable information about its quality and how it should be prepared. There are some points that should be clarified in order the manuscript to provide more proper information and also to meet the standards of the journal offering also an adding value to the readers.
Response: Many thanks for appreciating this opportunity to carefully evaluate our manuscript and these suggestions are quite instructive for this work. According to the reviewer’s comments, we carefully revised the manuscript and made the point-to-point response as below.
Specific comments:
Q1: Introduction--> Could you please refer which is the optimal texture for Sichuan-style braised beef according to preferences of consumers and if deviations from that texture are observed in the two products you refer?
Response: Thank you very much for your scientific suggestion. Due to incomplete description at the very start, we have added detailed discussion on the optimal texture of braised beef in the Introduction section of our revised manuscript. In addition, we have supplemented stewing times required for different types of braised beefs in the Conclusion section of the revised manuscript, and then copy the relevant content here for your check:
“Therefore, considering the texture properties, when Sichuan-style pre-braised beef bought by consumers is stewed with other ingredients for about 30 min, and the corresponding its stewing time is 60 min in industrialized production processes. When consumers only give simple heating to eat Sichuan-style pre-braised beef product, stewing time for 120 or 150 min can be considered in industrialized production processes.”
Q2: 2.1 the cattle slaughtered --> the slaughtered cattle.
Response: Thank you so much for this kind reminding. We have carefully changed " the cattle slaughtered " to " the slaughtered cattle" in the revised manuscript.
Q3: 2.2. specify the mass (gr) of each portion. In addition the percentages where do they refer to? final volume? final mass?
Response: We thank the reviewer for the very professional question. The added amounts of each auxiliary ingredient were expressed as a percentage of meat mass, and a short description have been supplemented in our revised manuscript, and then copy the relevant content here for your check:
“the added amount of each auxiliary ingredient was expressed as a percentage of meat weight”
Q4: 2.3-2.12 kindly describe briefly the methods you use and the initial volume or mass of the ingredient you use. In addition specify how many replicated did you apply. In 2.12 specify also how you conduct cluster analysis?
Response: Thank you very much for your scientific and rigorous reminders. We have supplemented the information in section 2.3-2.12. Additionally, the description of cluster analysis has been added in 2.12. The whole experiments were repeated 3 times at different times.
Q5: Table 1 specify the units if applicable in each trait. Please set also the superscripts a,b,c ... starting from the value with the lowest measure (the same in all tables).
Response: We appreciate the reviewer for the very instructive suggestion. We have carefully added the units of texture properties in Table 1 and modified the superscript order of each table.
Q6: Conclusions--> Authors suggest a stewing time 90 min or 60 min according to the type of product. Do the respective texture traits falls into consumers' expectations? Authors could make such a link even with previous reported results to justify stronger the results of their study.
Response: Thank you very much for your scientific and rigorous reminders. We are very sorry for the incomplete description. We have supplemented the information in the Conclusion section of the revised manuscript. In addition, we have added relevant references [Modzelewska-Kapituła, M.; Tkacz, K.; Nogalski, Z. The influence of muscle, ageing and thermal treat-ment method on the quality of cooked beef. J Food Sci Tech Mys. 2022, 59, (1), 123-132.] and [Li, J.; Han, D.; Huang, F.; Zhang, C. Effect of reheating methods on eating quality, oxidation and flavor characteristics of Braised beef with potatoes dish. Int J Gastronomy Food S. 2023, 31, 100659.] to describe consumers' expectations about the texture traits of braised beef in the revised manuscript.
Round 2
Reviewer 3 Report
Authors have improved their text considering all my reported comments. The manuscript offers valuable information and meet the standards of the journal. There are some minor points that authors should pay attention and further elaborate before any further publication. Kindly see bellow the specific points.
-and the corresponding its stewing time-->and its corresponding stewing time
-in a 4 °C workshop--> what do you mean workshop? specify
-2.9. According to your description you just specified only MDA and not total TBARS. Therefore, if my assumption is wright, please specify exactly in the subtitle that you refer toMDA and not TBARS
-2.11. specify the subroutine of R that you use for cluster analysis (which package?). Also, you stated that you repeated 3 times at different times. At which different times? Didi you also account for repeated measures in your analysis?
Author Response
Authors have improved their text considering all my reported comments. The manuscript offers valuable information and meet the standards of the journal. There are some minor points that authors should pay attention and further elaborate before any further publication. Kindly see below the specific points.
Response: Thank you very much for the reviewer’s approval of our revised manuscript. Your recognition and suggestions have given us great encouragement. All requested revisions based on the professional comments from the reviewer have been carried out.
Specific comments:
Q1: -and the corresponding its stewing time-->and its corresponding stewing time
Response: Thank you very much for your scientific and rigorous reminders. The “and the corresponding its stewing time” has been replaced by the “and its corresponding stewing time” in the revised manuscript.
Q2: -in a 4 °C workshop--> what do you mean workshop? Specify
Response: Thank you very much for your very professional question, we are very sorry for the unclear description. The “workshop” has been replaced by the “cold storage room” in section 2.1 of the revised manuscript.
Q3: -2.9. According to your description you just specified only MDA and not total TBARS. Therefore, if my assumption is wright, please specify exactly in the subtitle that you refer to MDA and not TBARS
Response: Thank you very much for your scientific advice, we have supplemented the describe in section 2.9 of the revised manuscript.
Q4: -2.11. specify the subroutine of R that you use for cluster analysis (which package?). Also, you stated that you repeated 3 times at different times. At which different times? Did you also account for repeated measures in your analysis?
Response: Thank you very much for your question. We deeply regret that the description is not clear in our previous manuscript. The heat map package was applied to the process of hierarchical cluster analysis, and we have added the specific information in section 2.11 of the revised manuscript. The experiments were repeated 3 times in a row on the same day while at different points in time, namely immediately repeating under the same conditions after completing the first experiment. In the analysis process, the average plus standard error of mean was used to represent the results after repeated measurements, as shown in Table 1 and Table 2.